# SALARECON connects the Atlantic salmon genome to growth and feed efficiency

Maksim Zakhartsev[1,2☉], Filip Rotnes [1,2☉], Marie Gulla [1,2☉], Ove Øyås [1,2☉], Jesse C. J. van Dam[3], Maria Suarez-Diez [3], Fabian Grammes [2], Róbert Anton Hafþórsson[2], Wout van Helvoirt [2], Jasper J. Koehorst [3], Peter J. Schaap[3], Yang Jin [2], Liv Torunn Mydland [2], Arne B. Gjuvsland [2], Simen R. Sandve[2], Vitor A. P. Martins dos Santos[3], Jon Olav Vik [1,2]*

1 Faculty of Chemistry, Biotechnology and Food Science, Norwegian University of Life Sciences (NMBU), Ås, Norway, 2 Faculty of Biosciences, Norwegian University of Life Sciences (NMBU), Ås, Norway, 3 Laboratory of Systems and Synthetic Biology, Wageningen University & Research (WUR), Wageningen, The Netherlands

☉ These authors contributed equally to this work.
* jon.vik@nmbu.no

**Data Availability Statement:** The model and scripts that reproduce our results can be found at gitlab.com/digisal/salarecon.

## Abstract

Atlantic salmon (*Salmo salar*) is the most valuable farmed fish globally and there is much interest in optimizing its genetics and rearing conditions for growth and feed efficiency. Marine feed ingredients must be replaced to meet global demand, with challenges for fish health and sustainability. Metabolic models can address this by connecting genomes to metabolism, which converts nutrients in the feed to energy and biomass, but such models are currently not available for major aquaculture species such as salmon. We present SAL-ARECON, a model focusing on energy, amino acid, and nucleotide metabolism that links the Atlantic salmon genome to metabolic fluxes and growth. It performs well in standardized tests and captures expected metabolic (in)capabilities. We show that it can explain observed hypoxic growth in terms of metabolic fluxes and apply it to aquaculture by simulating growth with commercial feed ingredients. Predicted limiting amino acids and feed efficiencies agree with data, and the model suggests that marine feed efficiency can be achieved by supplementing a few amino acids to plant- and insect-based feeds. SALARECON is a high-quality model that makes it possible to simulate Atlantic salmon metabolism and growth. It can be used to explain Atlantic salmon physiology and address key challenges in aquaculture such as development of sustainable feeds.

## Author summary

Atlantic salmon aquaculture generates billions of euros annually, but faces challenges of sustainability. Salmon are carnivores by nature, and fish oil and fish meal have become scarce resources in fish feed production. Novel, sustainable feedstuffs are being trialed hand in hand with studies of the genetics of growth and feed efficiency. This calls for a mathematical-biological framework to integrate data with understanding of the effects of novel feeds on salmon physiology and its interplay with genetics. We have developed the

**Funding:** This work was funded by the Research Council of Norway (https://www.forskningsradet.no/en/) grant 248792 (DigiSal) with support from grant 248810 (Centre for Digital Life Norway). The grant was awarded to JOV. The funders had no role in study design, data collection and analysis, decision to publish, or preparation of the manuscript.

SALARECON model of the core salmon metabolic reaction network, linking its genome to metabolic fluxes and growth. Computational analyses show good agreement with observed growth, amino acid limitations, and feed efficiencies, illustrating the potential for *in silico* studies of potential feed mixtures. In particular, *in silico* screening of possible diets will enable more efficient animal experiments with improved knowledge gain. We have adopted best practices for test-driven development, virtual experiments to assay metabolic capabilities, revision control, and FAIR data and model management. This facilitates fast, collaborative, reliable development of the model for future applications in sustainable production biology.

## Introduction

Salmonid aquaculture has grown in volume and economic importance over the past several decades, and Atlantic salmon (*Salmo salar*) has become the world's most valuable fish commodity [1]. This is largely thanks to selective breeding, which has improved both growth rate and feed efficiency [2]. The increase in fish farming has also increased demand for feed, and insufficient marine resources has led to a switch to plant-based ingredients [3]. This has reduced production costs and exploitation of fish stocks, but salmon are not adapted to eating plants and current plant-based feeds have a negative impact on fish health and the environment [4, 5]. Also, plant-based feeds are complex, the ingredient market is fluctuating, and feeding trials are demanding. Thus, developing feeds that minimize cost and environmental impact while providing necessary nutrients to the fish is an important challenge [6].

The metabolic network of a cell or organism converts nutrients that are present in the environment to the energy and building blocks that are required to live and grow. It consists of metabolites that are interconverted by metabolic reactions, most of which are catalyzed by enzymes that are encoded by the genome, and it can be translated to a metabolic model, which allows mathematical analysis of network functionality through methods such as flux balance analysis (FBA) [7]. Specifically, metabolic models allow prediction of growth and metabolic fluxes (steady-state reaction rates) that are linked to the genome through logical gene-protein-reaction (GPR) associations, making them promising tools for addressing challenges in aquaculture such as breeding for feed efficiency and sustainable feed development [8]. Large databases of metabolic reactions and models [9–11] and methods for metabolic network reconstruction from annotated genomes [12, 13] have made such models available for organisms ranging from microbes to animals [14]. However, there are still very few metabolic models of fish available [15–18] and none of Atlantic salmon or other important farmed fish species.

Here, we present SALARECON: a metabolic model built from the Atlantic salmon genome [19] that predicts growth and metabolic fluxes. It has been manually curated to ensure flux consistency and focuses on energy, amino acid, and nucleotide metabolism. SALARECON is a high-quality model according to community-standardized tests, and it captures expected metabolic (in)capabilities such as amino acid essentiality. Using oxygen-limited growth under hypoxia as an example, we show that model predictions can explain salmon physiology in terms of metabolic fluxes that are, in turn, tied to genes and pathways. Furthermore, we demonstrate an important application for aquaculture by predicting growth-limiting amino acids and feed efficiencies for commercial feed ingredients in agreement with data.

## Methods

### Building the metabolic model

We manually built a draft model focusing on Atlantic salmon energy, amino acid, and nucleotide metabolism using the genome [19] with annotations from KEGG [11] and the software Insilico Discovery (Insilico Biotechnology, Stuttgart, Germany). Pathways were added or edited one by one with information about reactions obtained from databases and literature (S1 Fig). After adding or editing a pathway, the energy and redox balances and topological properties of the model, e.g. flux consistency, were checked. Based on the results from these analyses, the pathway was either kept or modified. Before final acceptance of a pathway, FBA was performed to ensure that the model was able to predict growth and metabolic fluxes. We used WoLF PSORT [20] with default settings through SAPP [21] to assign metabolites and reactions to six different compartments (cytosol, mitochondrion, inner mitochondrial membrane, extracellular environment, peroxisome, and nucleus). Exchange reactions were added to allow metabolite import (negative flux) and export (positive flux).

After finishing the draft model, we converted the model to the BiGG [10] namespace and used COBRApy [22] to iteratively curate it. We added and removed metabolites, reactions, and genes, mapped genes to reactions using AutoKEGGRec [23], and added a salmon-specific biomass reaction. We also added annotations from MetaNetX [9], KEGG [11], UniProt [24], and NCBI [25]. To infer GPR associations for reactions, we mapped Atlantic salmon genes to human homologs and copied GPR associations from the most recent human model [26]. If no GPR association could be inferred for a reaction, we used an OR relation between genes mapped to that reaction. To build the biomass reaction, we estimated the fractional composition of macromolecules in 1 g dry weight biomass (gDW) from Atlantic salmon whole-body composition [27]. We mapped macromolecules to metabolites and estimated the fractional composition of amino acids in proteins and nucleoside triphosphates in nucleic acids from proteome and genome sequences [19], respectively. We finalized the model by alternating semi-automated annotation and curation with quality evaluation (as decribed below), iterating until we saw no further opportunities to improve the model without expanding its scope beyond energy, amino acid, and nucleotide metabolism. The final model was exported to Systems Biology Markup Language (SBML) format [28].

### Evaluating the quality of the metabolic model

First, we compared the reaction contents in SALARECON to other models of multicellular eukaryotes available in the BiGG [10] namespace (*Danio rerio* [17], *Mus musculus* [29], *Cricetulus griseus* [30], *Homo sapiens* [26], and *Phaeodactylum tricornutum* [31]). Considering only intracellular metabolic reactions in compartments shared by all models (cytoplasm, mitochondrion, and peroxisome), we clustered the models based on their reaction contents using all suitable dissimilarity measures (16) and agglomerative hierarchical clustering methods (5) available through SciPy [32]. For each measure and method, we evaluated the resulting dendrogram by computing the cophenetic correlation coefficient (CCC) [33]:

$$\text{CCC} = \frac{\sum_{i<j}(x(i,j) - \bar{x})(t(i,j) - \bar{t})}{\sqrt{(\sum_{i<j}(x(i,j) - \bar{x})^2)(\sum_{i<j}(t(i,j) - \bar{t})^2)}}, \tag{1}$$

where $x(i,j)$ and $t(i,j)$ are Euclidean and dendrogrammatic distance between observations $i$ and $j$, respectively, with averages $\bar{x}$ and $\bar{y}$. The CCC indicates how well the dendrogram preserves pairwise dissimilarities.

Second, we tested SALARECON's consistency and annotation using the community standard MEMOTE [34] and its metabolic (in)capabilities using tasks defined for mammalian cells [35]. We adapted tasks to Atlantic salmon by moving metabolites from compartments not included in SALARECON to the cytoplasm and by modifying the expected outcomes of amino acid synthesis tests to match known essentiality [27, 36].

Third, we used the model to predict growth in the absence of individual amino acids. We allowed both uptake and secretion of all extracellular metabolites, disabled uptake of each amino acid separately, and maximized growth rate using FBA. Amino acids were classified as essential if they were required for growth and non-essential otherwise, and the predicted essentiality was compared to experimental data [27, 36].

Finally, we evaluated the ability of SALARECON to capture fish-specific metabolism by comparing metabolite uptake and secretion to the most recent human model [26]. Specifically, we enumerated minimal growth-supporting uptake and secretion sets for both models using scalable metabolic pathway analysis [37]. We required non-zero growth rate, uptake of oxygen and essential amino acids, and secretion of carbon dioxide as well as ammonia, urea, or urate. We enumerated minimal uptake sets first and then allowed uptake of all metabolites found in uptake sets before enumerating minimal secretion sets.

## Analyzing oxygen-limited growth

We used parsimonious FBA (pFBA) [38] to find maximal growth rates and minimal flux distributions for 1,000 randomized conditions and 50 logarithmically spaced oxygen uptake rates in the range $r \in (0, r_{max})$ where $r$ is oxygen uptake rate and $r_{max}$ is the minimal oxygen uptake rate at maximal growth. For each condition, we uniformly sampled random ratios (1–100) of nutrients in a minimal feed (essential amino acids and choline) that were used as coefficients in a boundary reaction representing feed uptake. We always normalized feed uptake to the same total mass (g $gDW^{-1}$ $h^{-1}$) to ensure that conditions were comparable. The absolute value was selected to be large enough to ensure feed uptake was not limiting but otherwise arbitrary as only relative predictions were needed. We allowed unlimited uptake of phosphate and disabled all other uptakes as well as secretion of feed nutrients. We did not allow uptake of any other compounds than essential amino acids, choline, phosphate, and oxygen under any condition.

To account for uncertainty in relative flux capacities and ensure that no single set of reactions was always growth-limiting, we also sampled random bounds for all reactions for each condition. The flux bound $b$ of an enzymatic reaction is determined by the turnover number $k_{cat}$ and total enzyme concentration [E]:

$$b = k_{cat}[\mathrm{E}]. \tag{2}$$

Approximately lognormal distributions have been observed for both $k_{cat}$ [39] and [E] [40], and the product of two lognormal random variables is also lognormal. We therefore sampled $b$ from a lognormal distribution with mean 0 and standard deviation 2 for the natural logarithm of $b$. We kept the original reaction reversibilities and sampled bounds for reversible reactions separately for each direction.

For each oxygen uptake rate, we computed mean growth rate with 95% confidence band from bootstrapping with 1,000 samples. We fit the means to experimental data [41–44] by assuming a simple piecewise linear relationship between water oxygen saturation ($x$) and

relative oxygen uptake rate:

$$\frac{r}{r_{\max}} = \begin{cases} 0 & x \leq x_0 \\ \dfrac{x - x_0}{x_1 - x_0} & x \in (x_0, x_1), \\ 1 & x \geq x_1 \end{cases}$$

(3)

where $x_0$ and $x_1$ are the oxygen saturations at which the relative growth rate is 0 and 1, respectively. We estimated $x_0$ and $x_1$ by least-squares fitting of

$$\frac{\mu}{\mu_{\max}} = f\left(\frac{r}{r_{\max}}\right),$$

(4)

where $\mu$ is growth rate, $\mu_{\max}$ is maximal growth rate when oxygen is not limiting, and $f$ is a function that linearly interpolates the metabolic model predictions. We also fit a logistic model with asymptotes -1 and 1,

$$\frac{\mu}{\mu_{\max}} = \frac{2}{1 + e^{k(x_0 - x)}} - 1,$$

(5)

where $k$ is the logistic growth rate, and a Monod model extended with an $x$-intercept,

$$\frac{\mu}{\mu_{\max}} = \frac{x - x_0}{K_s + x - x_0},$$

(6)

where $K_s + x_0$ is the saturation at which $\mu = \frac{1}{2}\mu_{\max}$.

To test the effect of random sampling on predictions and parameter estimates, we also repeated the analysis above with 100 randomly sampled feeds and default flux bounds as well as with a fish meal feed (Table 1) and default flux bounds. The fish meal feed includes non-

**Table 1. Amino acid compositions of feed ingredients.** Mass percentage of each amino acid relative to total mass of amino acids in feed ingredients used in simulations [57].

| Amino acid | Fish meal | Soybean meal | Insect meal |
|---|---:|---:|---:|
| Ala | 6.82 | 4.43 | 7.05 |
| Arg | 7.19 | 7.54 | 5.34 |
| Asn/Asp | 10.02 | 11.87 | 10.07 |
| Cys | 0.93 | 1.74 | 0.62 |
| Gln/Glu | 13.98 | 18.74 | 11.12 |
| Gly | 6.88 | 4.19 | 6.67 |
| His | 2.62 | 2.69 | 3.32 |
| Ile | 4.64 | 4.61 | 4.86 |
| Leu | 7.91 | 8.02 | 7.76 |
| Lys | 8.31 | 6.44 | 6.19 |
| Met | 3.07 | 1.45 | 2.06 |
| Phe | 4.29 | 5.22 | 4.31 |
| Pro | 4.45 | 5.08 | 6.39 |
| Ser | 4.29 | 4.13 | 4.71 |
| Thr | 4.57 | 3.67 | 4.29 |
| Trp | 1.13 | 1.58 | 1.61 |
| Tyr | 3.40 | 3.60 | 6.85 |
| Val | 5.48 | 5.00 | 6.79 |

essential amino acids that were not present in the randomly sampled minimal feeds and we also allowed unlimited uptake of the lipid precursor choline. This gave a non-zero maximal growth rate in the absence of oxygen, which we subtracted from predicted growth rates in the presence of oxygen to get purely aerobic growth rates suitable for comparison to the other growth rate predictions. We identified limiting reactions with and without random sampling of feeds and flux bounds by comparing predicted fluxes to their corresponding non-zero lower and upper flux bounds. A reaction was identified as limiting if the distance between predicted flux and one of its non-zero flux bounds was within the numerical tolerance of the solver.

To identify reaction contributions to oxygen-limited growth, we took the absolute value of the pFBA fluxes (with randomly sampled feeds and flux bounds), normalized each flux by its maximum value within each condition, and used Ward's minimum variance method to cluster the resulting absolute relative fluxes by Euclidean distance. We mapped reactions from the top eight clusters to genes and used g:Profiler [45] to identify enriched pathways from KEGG [11]. We used the genes in the model as background, considered pathways with adjusted $p \leq 0.05$ to be enriched, and discarded pathways outside the model's scope (xenobiotics and drug metabolism).

### Predicting growth-limiting amino acids in feeds

We obtained ratios of amino acids in three commercial feed ingredients: fish, soybean, and black soldier fly larvae meal (Table 1). For each feed, these ratios were used as coefficients for amino acids in a boundary reaction representing feed consumption. Mass was divided equally between amino acids that were combined in the feed formulation (Asn/Asp and Gln/Glu). For each feed ingredient, we deactivated import of amino acids via other boundary reactions, fixed the growth rate to the same value (arbitrary, as we were interested in generated biomass relative to consumed feed), and normalized feed uptake to the same total mass (g gDW$^{-1}$ h$^{-1}$) before minimizing feed uptake flux. To simulate growth limitations from protein synthesis rather than energy generation, we also allowed unlimited uptake of glucose. This is supported by evidence that reducing feed amino acid levels has a negative effect on feed intake regardless of dietary energy level [46]. We multiplied molecular mass with reduced cost in the optimal solution for each amino acid exchange reaction and identified the one with largest negative value as limiting [47]. To supplement the feed with the limiting amino acid, we set the bounds of its exchange reaction to only allow import, and we penalized supplementation by adding the exchange reaction to the objective with coefficient equal to molecular mass (S2 Fig). We repeated the steps above until all limiting amino acids had been found for each feed.

## Results

We built a metabolic model of Atlantic salmon (SALARECON) from its genome [19], metabolic reaction and model databases, and literature (Fig 1). The model focuses on energy, amino acid, and nucleotide metabolism and covers 1,133 genes, which amounts to 2% of the 47,329 annotated genes in the genome and 50% of the 2,281 Atlantic salmon genes that are associated with metabolic reactions in KEGG [11]. The genes are mapped through gene-protein-reaction (GPR) associations to a metabolic network of 718 reactions and 530 metabolites (Fig 2a) with node degree distributions that are typical for metabolic and other biological networks [48] (S3 Fig). Reactions and metabolites are divided between six compartments: cytosol, mitochondrion, inner mitochondrial membrane, extracellular environment, peroxisome, and nucleus (Fig 2b). The compartments are connected by 175 transport reactions that allow metabolite exchange through the cytosol, and 86 boundary reactions allow metabolites to move in and out of the system through the extracellular environment. There are 357 unique

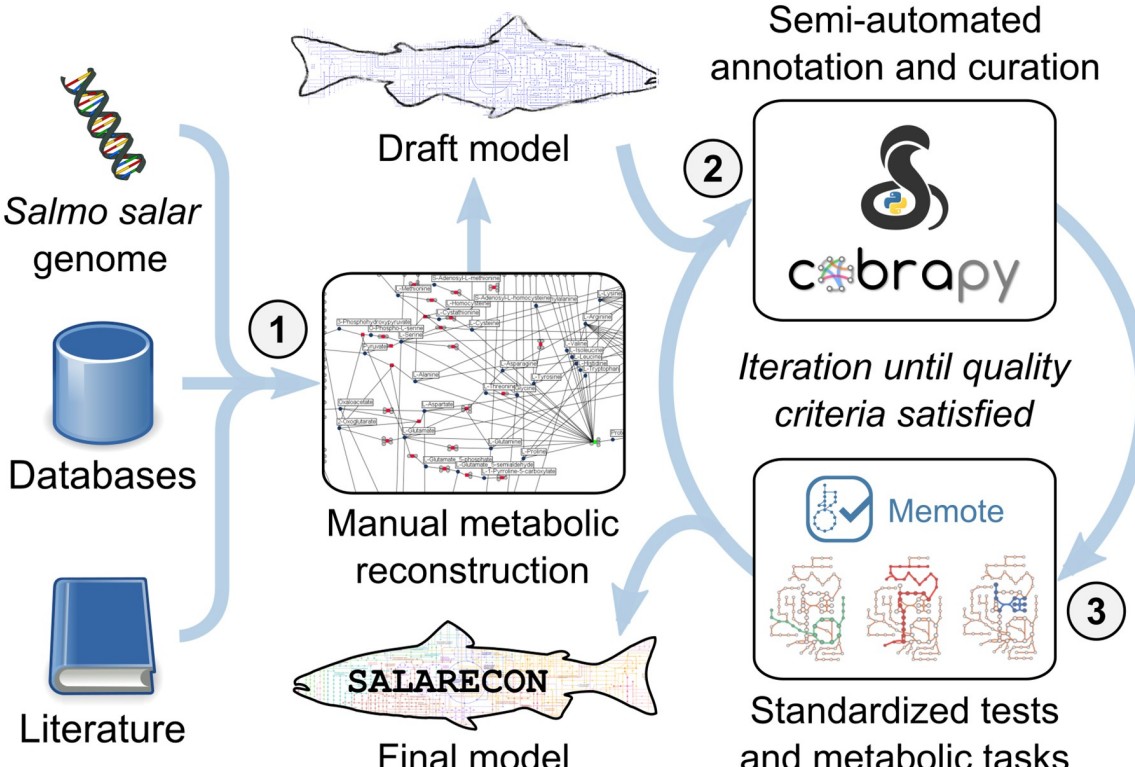

**Fig 1. Model construction.** SALARECON was built from the annotated Atlantic salmon genome, metabolic reaction and model databases, and literature. The procedure involved (1) manual metabolic network reconstruction using Insilico Discovery (Insilico Biotechnology, Stuttgart, Germany), (2) semi-automated annotation and curation using COBRApy [22], and (3) quality evaluation using the standardized metabolic model testing tool MEMOTE [34] and metabolic tasks [35]. Steps 2 and 3 were iterated until quality criteria were satisfied. Illustration of metabolic tasks from Richelle et al. [35].

metabolites when those occurring in multiple compartments are counted once. A salmon-specific biomass reaction based on whole-body composition [27] allows growth rate prediction by accounting for production of the proteins, lipids, carbohydrates and nucleic acids that consitute biomass from metabolites supplied by the metabolic network (Fig 2c).

To investigate whether SALARECON is likely to be an accurate representation of Atlantic salmon metabolism, we first compared it to the only existing high-quality metabolic model of a fish as well as all models of multicellular eukaryotes currently available in the BiGG database [10] (Fig 3a, S4 and S5 Figs). Specifically, we hierarchically clustered the reaction contents of SALARECON and models of zebrafish (*Danio rerio*) [17], mouse (*Mus musculus*) [29], chinese hamster ovary (CHO; *Cricetulus griseus*) [30], human (*Homo sapiens*) [26], and the diatom *Phaeodactylum tricornutum* [31]. We combined 16 different dissimilarity measures with five different methods for agglomerative hierarchical clustering, and we evaluated the agreement between the resulting dendrograms and the underlying dissimilarities using the cophenetic correlation coefficient (CCC) [33] (S4 Fig). Across measures and methods, we found that models tended to cluster by phylogeny with fish and, to some extent, mammals forming distinct clusters and the diatom being an outlier. As shown in S4 Fig, salmon and zebrafish formed a cluster in 57/80 trees (71%), the mammals formed a cluster in 36/80 trees (45%), and the diatom was an outlier in 69/80 trees (86%). This is largely consistent with the hypothesis that the models capture organism-specific metabolism, suggesting that SALARECON captures

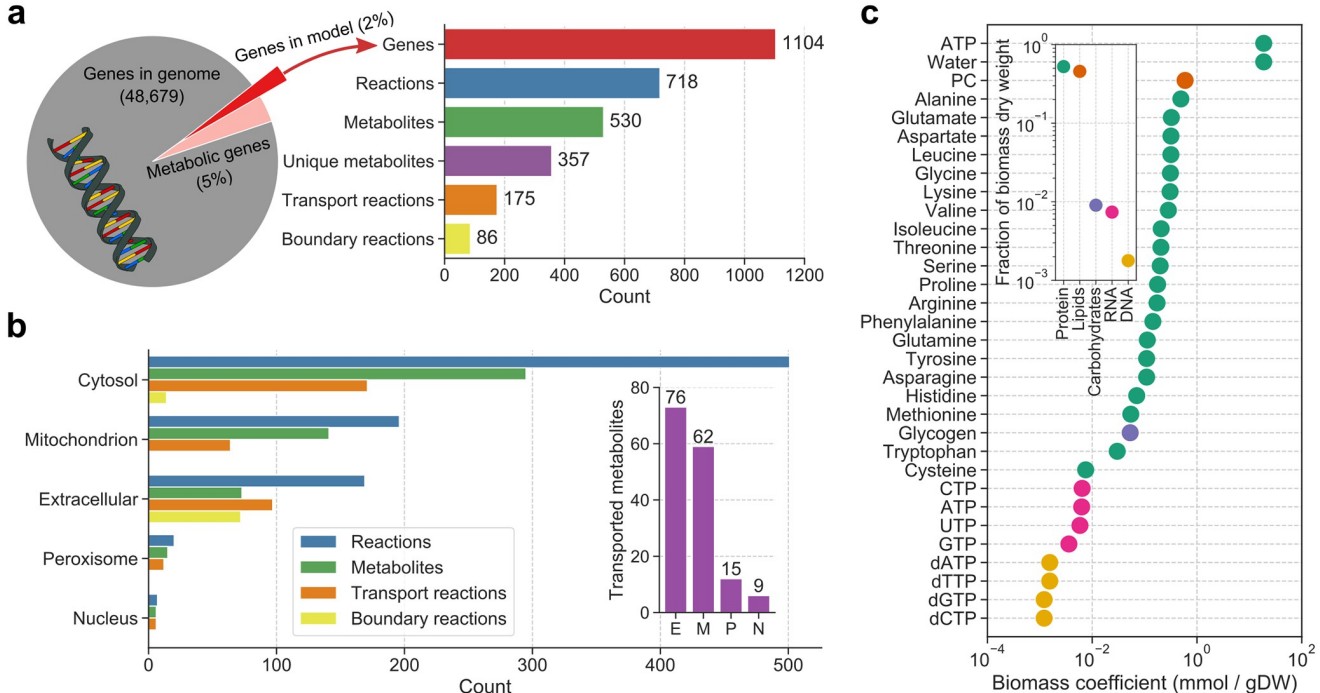

**Fig 2. Model contents.** (a) SALARECON contains 1,133 genes (2% of all genes and 50% of Atlantic salmon genes mapped to reactions in KEGG [11]), 718 reactions (175 transporting metabolites between compartments and 86 exchanging metabolites with the extracellular environment), and 530 metabolites (357 when metabolites occuring in multiple compartments are only counted once). (b) Metabolites and reactions are divided between five compartments (mitochondrion includes the inner mitochondrial membrane). Transport reactions are counted multiple times (once for each compartment of exhanged metabolites). Boundary reactions in cytosol are sink or demand reactions [12]. The inset shows how many unique metabolites can be transported between the cytosol and the other compartments (indicated by their initials). (c) Biomass composition of Atlantic salmon estimated from measured whole-body composition [27]. The inset summarizes each class of macromolecules. Carbohydrates and lipids are represented by glycogen and phosphatidylcholine (PC), respectively. ATP serves both as energy for protein synthesis and as a building block in RNA synthesis.

salmon- or at least fish-specific metabolism. However, there are also significant discrepancies between trees built with different measures and methods, and it is important to note that the clusters likely reflect large differences in model scope as well as organism specificity (S5 Fig). Fig 3a shows the tree obtained for Jaccard distance, which is the most common metric for measuring metabolic model dissimilarity [49], with the "average" method (CCC = 0.95).

SALARECON performed well in community-standardized MEMOTE tests [34], which evaluate model consistency and annotation (Fig 3b). It achieved an overall MEMOTE score of 96% (best possible score is 100%) with subscores of 100% for Systems Biology Ontology (SBO) annotation, 98% for model consistency, 94% for metabolite annotation, 87% for reaction annotation, and 71% for gene annotation. We also evaluated the ability of SALARECON to perform 210 metabolic tasks grouped into seven metabolic systems (Fig 3c) and 73 metabolic subsystems (S6 Fig). These tasks were originally defined for mammalian cells [35] but we changed the expected outcomes of amino acid synthesis tests to match known essentiality in Atlantic salmon [27, 36]. SALARECON correctly captured all expected metabolic (in)capabilities for the three metabolic systems within the scope of the model (energy, amino acid, and nucleotide metabolism). It also succeeded in 44% of vitamin and cofactor tasks, 43% of carbohydrate tasks, and 15% of lipid tasks, reflecting the fact that these parts of metabolism are simplified in the model. The only system completely outside the scope of SALARECON was glycan metabolism, in which no tasks were successfully performed. In total, SALARECON

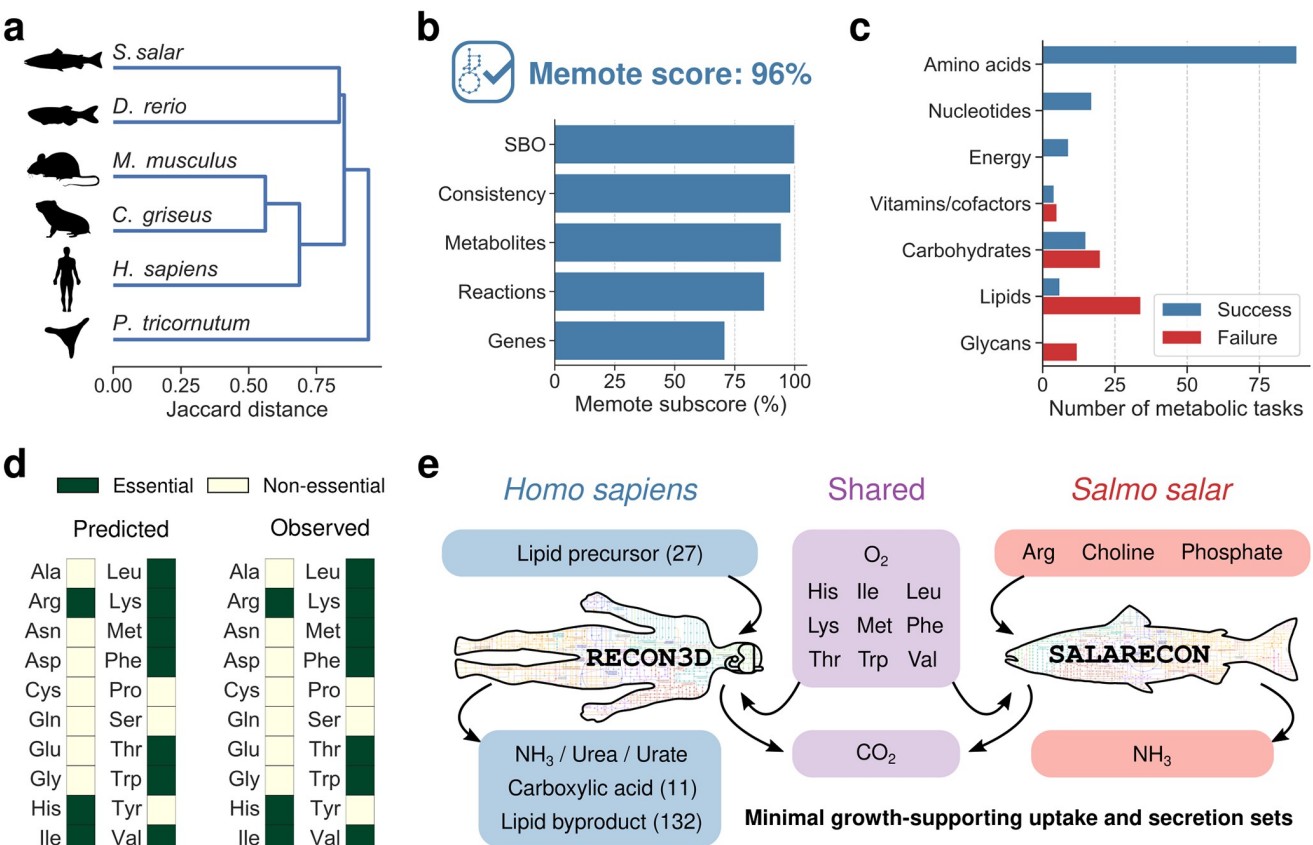

**Fig 3. Model quality evaluation.** (a) Hierarchical clustering of SALARECON and metabolic models of other multicellular eukaryotes based on Jaccard distance between reaction contents and the "average" method. Atlantic salmon (*Salmo salar*) is closer to zebrafish (*Danio rerio*) [17] than mouse (*Mus musculus*) [29], chinese hamster ovary (CHO; *Cricetulus griseus*) [30], human (*Homo sapiens*) [26], and the diatom *Phaeodactylum tricornutum* [31]. (b) Model score and subscores from MEMOTE [34]. Subscores evaluate Systems Biology Ontology (SBO) annotation, model consistency, and database mappings for metabolites, reactions, and genes. (c) Ability of SALARECON to perform metabolic tasks [35]. Tasks are grouped by metabolic system and classified as successful if model predictions reflected expected metabolic (in)capabilities. (d) Essential amino acids predicted by SALARECON match observations [27, 36]. (e) Minimal growth-supporting sets of metabolite uptakes and secretions for RECON3D [26] and SALARECON. Arrows indicate uptake and secretion. Metabolites that are only used or produced by human or salmon are indicated by blue and red, respectively, and metabolites that are used or produced by both are indicated in purple. Uptake of oxygen and essential amino acids was required, as well as secretion of carbon dioxide and ammonia, urea, or urate. Ammonia secretion can be replaced by urea or urate secretion in human but not in salmon. The number of alternative metabolites is given in parentheses where applicable.

succeeded in 66% of all metabolic tasks, notably all tasks related to amino acid essentiality (Fig 3d).

Finally, to test the ability of SALARECON to capture basic fish physiology, we compared it to one of the latest human models, RECON3D [26], by computing minimal sets of metabolite uptakes and secretions that allow growth [37] (Fig 3e). In addition to oxygen and essential amino acids, SALARECON required uptake of choline, a lipid precursor, and phosphate, an essential nutrient for fish that is supplemented in salmon feeds [50]. The only secretions needed to support growth were carbon dioxide and ammonia. Notably, we found that secretion of urea was also possible, but not sufficient to support growth without secretion of ammonia. In line with this, ammonia is the major nitrogenous waste product in fish and urea is a comparatively minor contributor [51]. RECON3D is much larger than SALARECON and therefore allowed for a wider range of lipid precursors (27 options). It also required secretion

of a carboxylic acid (11 options) and a lipid byproduct (132 options) in addition to carbon dioxide and a nitrogenous waste product. Urea is the major nitrogenous waste in mammals, but RECON3D could grow while secreting only ammonia or urate. In general, RECON3D captures a much larger space of possible growth-associated metabolic activities than SALARECON due to the large difference in model scope (S5 Fig). However, SALARECON specifically captures the key metabolic activities of a fish.

In our first application of SALARECON, we predicted oxygen-limited growth rates under hypoxia on a minimal feed containing essential amino acids and choline, using random sampling to account for uncertainty in feed nutrient ratios and flux capacities (Fig 4a and S7 Fig). Supporting the hypothesis that SALARECON captures fish-specific metabolism, we found that the major secretion products across all oxygen levels and sampled conditions were $CO_2$ and $NH_3$ (S8 Fig). Urea had the third highest secretion flux but this was much smaller than the secretion flux for $NH_3$, and the secretion fluxes of all other secreted metabolites combined was vanishingly small. Assuming that relative oxygen uptake rate is a linear function of water oxygen saturation (percent air saturation), we fit our predictions to experimental data [41–44] along with a logistic model and an extended Monod model (Fig 4b). The choice of a linear model for the metabolic fit was motivated by the fact that diffusive oxygen uptake in fish gills is governed by Fick's law and therefore proportional to the oxygen gradient [52]. Also, replacing the linear model by a Michaelis-Menten model would make the metabolic and Monod fits virtually identical because the Monod and Michaelis-Menten equations have the same form. We found that the metabolic, logistic, and extended Monod models fit the data about equally well ($R^2 \approx 0.6$) but they differed in their parameter estimates (Fig 4b). All the models estimated the minimal oxygen saturation required for growth, but the logistic estimate was low with high standard error ($x_0 = 0.11 \pm 0.16$) and the Monod fit was high with low standard error ($x_0 = 0.45 \pm 0.04$). The metabolic model gave an intermediate estimate and standard error ($x_0 = 0.31 \pm 0.10$), and it also allowed estimation of the minimal oxygen saturation required for *maximal* growth ($x_1 = 1.37 \pm 0.15$). The metabolic fit was closer than the two other fits to the expected relationship between water oxygen saturation and growth rate [52], both in terms of the shape of the fitted curve and the estimated parameter values. The SALARECON estimates were within one and two standard errors, respectively, of the values $x_0 \approx 0.3$ and $x_1 \leq 1.2$ suggested by Thorarensen et al. [52]. The logistic estimate was within two standard errors of the suggested $x_0$, but this confidence interval also included zero.

Repeating our oxygen-limited growth analysis with default rather than randomly sampled flux bounds, we found similar growth predictions and parameter estimates for 100 randomly sampled feeds as well as a feed based on fish meal (S9 Fig). However, random sampling of feeds allowed us to account for a much larger selection of potentially limiting reactions, showing that our results were robust to uncertainty in flux capacities as well as feed compositions (S10 Fig). With randomly sampled feeds and bounds, 310 different reactions were limiting in at least one solution, compared to 25 with the default bounds (for both randomly sampled and fish meal feeds). Growth was always limited by the flux capacities of internal reactions (and oxygen uptake) rather than by the feed uptake reaction.

In contrast to the simple growth models, SALARECON is mechanistic and makes it possible to explain predictions in terms of metabolic fluxes (Fig 4d). Assuming that organisms have generally evolved to grow as efficiently as possible, we used parsimonius flux balance analysis (pFBA) [38] to minimize overall flux through the metabolic network while requiring maximal growth rate for each randomly sampled condition and oxygen level. We identified eight clusters of reactions whose pFBA fluxes made distinct contributions to oxygen-limited growth (Fig 4e). Connecting clusters to the Atlantic salmon genome and databases through GPR

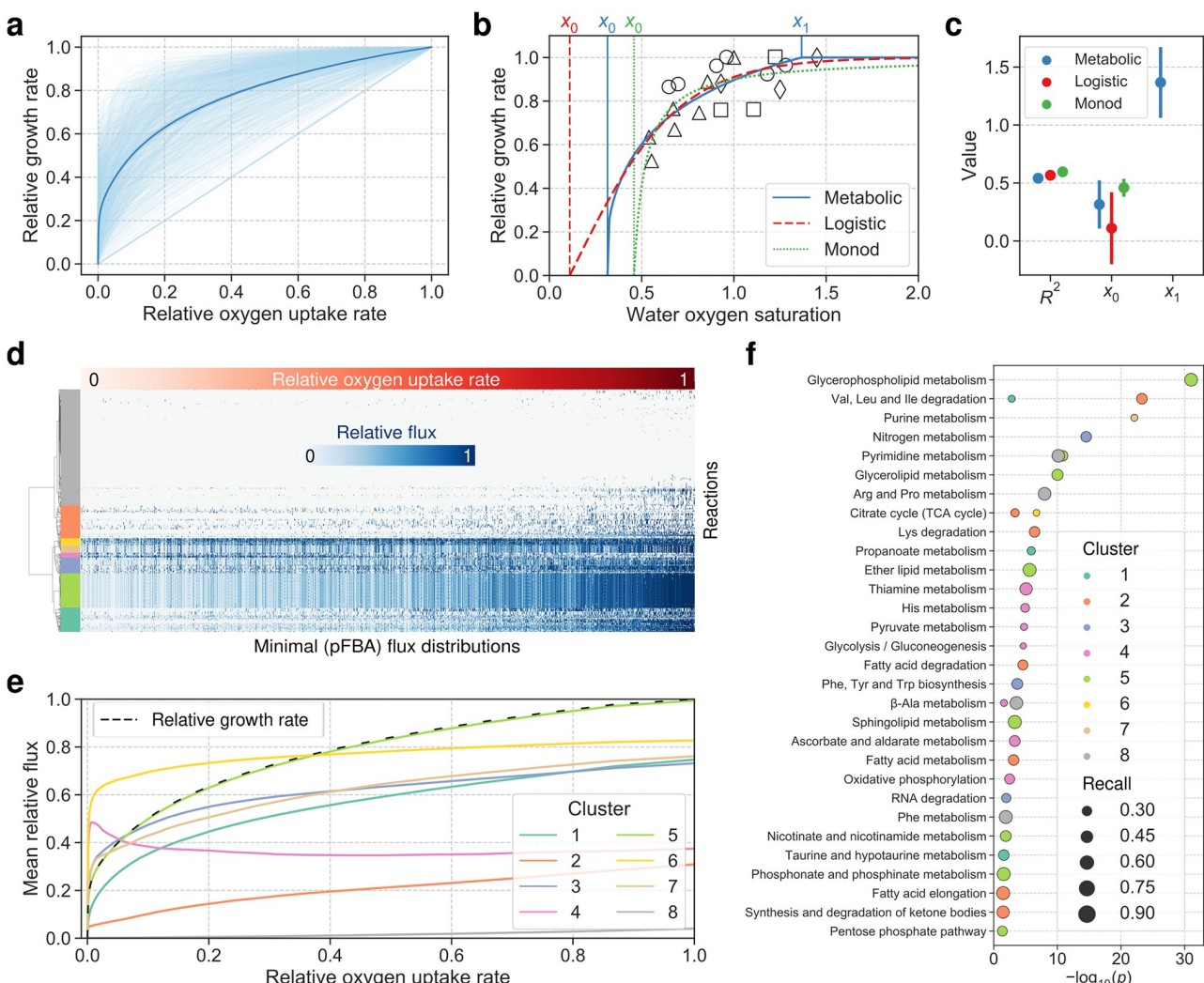

**Fig 4. Oxygen-limited growth analysis.** (a) SALARECON predictions of relative growth rate under oxygen limitation as a function of relative oxygen uptake rate. Feed composition and flux capacities were randomized 1,000 times (light blue) and the mean across conditions is shown with 95% confidence band from bootstrapping with 1,000 samples (dark blue). (b) Metabolic, logistic, and extended Monod model fits to experimental data from Berg and Danielsberg [41] (circles), Bergheim et al. [42] (triangles), Hosfeld et al. [43] (squares), and Hosfeld et al. [44] (diamonds). SALARECON predictions were fit by assuming a linear relationship between relative oxygen uptake rate and water oxygen saturation. (c) Coefficient of determination ($R^2$), minimal oxygen saturation required for growth ($x_0$), and minimal oxygen saturation required for *maximal* growth ($x_1$) from fitted models. Error bars indicate two standard errors of the estimates. (d) Minimal flux distributions for metabolic model predictions from parsimonious flux balance analysis (pFBA) [38]. Rows are reactions, columns are flux distributions sorted by relative oxygen uptake rate, and each cell shows absolute flux normalized by maximum value for each condition. Rows are clustered by Euclidean distance using Ward's minimum variance method and divided into eight clusters indicated by colors. (e) Mean absolute relative flux with 95% confidence bands from bootstrapping with 1,000 samples for the eight clusters. Relative growth rate is indicated by a dashed line. (f) Enrichment of metabolic pathways from KEGG [11] for the eight clusters with size reflecting the fraction of genes in each pathway that are found in a cluster (recall).

associations and their annotation, we found enriched metabolic pathways among the genes associated with each cluster (Fig 4f).

In one cluster, fluxes were perfectly correlated with relative growth rate, indicating that they contained reactions that were always necessary for growth. Indeed, this cluster was enriched in lipid metabolism, which directly produces a biomass precursor, and pathways related to NAD(P)H metabolism. The fluxes of two other clusters both increased rapidly at the

very lowest oxygen levels before plateauing at higher oxygen levels, in one case decreasing slightly after the initial increase. These clusters were enriched in pathways such as the tricarboxylic acid (TCA) cycle, glycolysis, oxidative phosphorylation, pyruvate, and thiamine metabolism, indicating that energy generation from glucose was maximized at low oxygen levels while other energy-generating pathways were activated at higher oxygen levels. Four of the five remaining clusters increased slightly less than the clusters enriched in energy generation from glucose at low oxygen levels but kept increasing at higher oxygen levels. These clusters were enriched in pathways related to metabolism of fatty acids and amino acids, suggesting that these compounds become important energy sources after saturation of glucose catabolism at low oxygen levels. Nitrogen metabolism, which includes amino acid biosynthesis and disposal of nitrogenous waste products, was also overrepresented. The final cluster consisted of reactions with no or very little flux, even at the highest oxygen levels, and was enriched in metabolism of pyrimidines, $\beta$-alanine, and essential amino acids.

Finally, to demonstrate the potential of SALARECON to address key challenges in aquaculture, we used it to predict growth-limiting amino acids and feed efficiencies for three commercial feed ingredients: fish, soybean, and insect meal (Table 1 and Fig 5a). For each feed ingredient, we iteratively identified and supplemented the most limiting amino acid until all

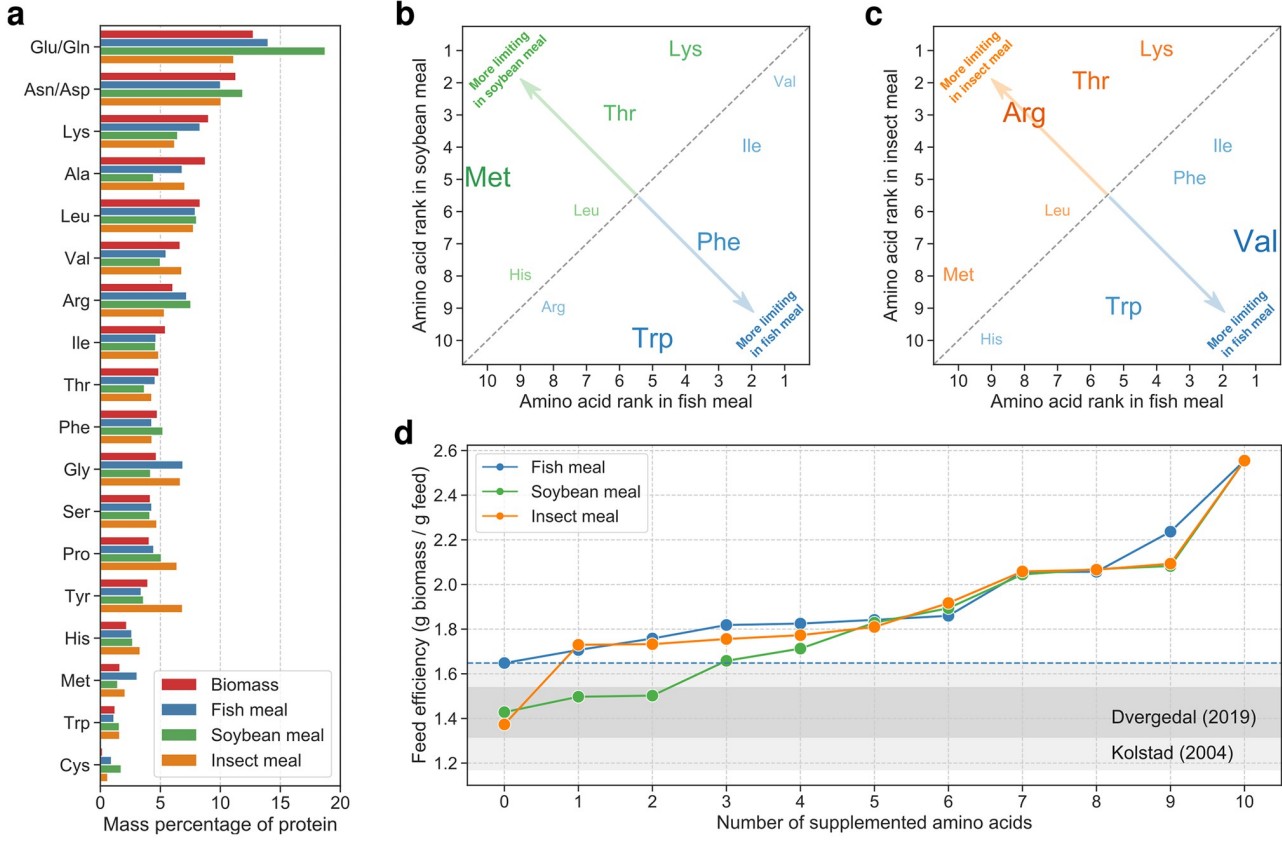

**Fig 5. Growth-limiting amino acids in commercial feed ingredients.** (a) Amino acid composition of SALARECON biomass, fish meal, soybean meal, and insect meal [57]. (b) Order of amino acid limitations in feed ingredients based on soybean and fish meal. Amino acids that are closer to the top left and bottom right corners were more limiting in soybean meal and fish meal, respectively, as indicated by size. (c) Order of amino acid limitations in feed ingredients based on insect and fish meal. Amino acids that are closer to the top left and bottom right corners were more limiting in insect meal and fish meal, respectively, as indicated by size and color. (d) Feed efficiency after successive supplementation of the most limiting amino acid for fish, soybean, and insect meal. The baseline feed efficiency of fish meal is indicated by a dashed blue line, and ranges observed by Kolstad et al. [53] and Dvergedal et al. [54] are highlighted in gray.

amino acid limitations had been lifted, computing feed efficiency at each iteration (S2 Fig). Comparing predicted limiting amino acids in fish meal to soybean and insect meal, we found that lysine and threonine were more limiting in both soybean and insect meal, methionine was more limiting in soybean meal, and arginine was more limiting in insect meal (Fig 5a and 5c, and S11 Fig). The feed efficiency predictions suggest that the baseline feed efficiency of fish meal can be achieved by supplementing one and three amino acids for soybean and insect meal, respectively (Fig 5d). For soybean meal, major increases in feed efficiency were predicted for lysine, threonine, and methionine supplementation, while lysine had the largest impact on insect meal (S11 Fig). The predictions from SALARECON agree well with expected baseline feed efficiencies [53, 54] as well as reports that lysine, methionine, threonine, and arginine are more limiting in plant-based feeds than in marine feeds [55, 56].

## Discussion

SALARECON is the first metabolic model of a production animal, bridging the gap between production and systems biology and initiating a framework for adapting Atlantic salmon breeding and nutrition strategies to modern feeds. By explicitly representing connections between metabolites, reactions, and genes, it connects the genome to metabolism and growth in a way that can be tuned to specific genetic and environmental contexts by integration of domain knowledge and experimental data [8]. Thus, SALARECON forms a transdiciplinary framework for diverse disciplines and data sets involved in Atlantic salmon research and aquaculture. Tools developed for constraint-based modeling of microbes and well-studied plants and animals can now be applied in production biology, providing a sharper lens through which to interpret omics data by requiring consistency with flux balances and other known constraints. This enables clearer analysis than classical multivariate statistics, which does not incorporate such mechanistic knowledge.

Although laborious and time-consuming, our bottom-up manual reconstruction of the Atlantic salmon metabolic network was necessary to make SALARECON a high-quality predictive model. Automatically built models work well for microbes but are still outperformed by models that are built by manual iteration, and reconstruction of eukaryotes is more challenging due to larger genomes, less knowledge, and compartmentalization [12, 13]. However, semi-automated annotation and curation combined with automated MEMOTE tests [34] and metabolic tasks [35] allowed faster iteration, and future reconstructions of related species [58] can benefit from our efforts by using SALARECON as a template. MEMOTE and metabolic tasks were instrumental in the development of SALARECON, and we highly recomend integrating testing in model development. Tests help catch mistakes that arise when modifying a model and do triple duty by specifying what it should be capable of, identifying broken functionality, and forming a basis for comparison with other models, e.g. new versions or models of different tissues or species. Clearly formulated tests also make the model more accessible to non-modelers, speaking the same language as nutritionists or physiologists. Such experts can point out missing or ill-formulated tests, which in turn contribute to improvement.

We have strived to make SALARECON an accurate model of Atlantic salmon metabolism and growth, but it does not aim to capture salmon physiology exhaustively or perfectly. It covers 2% of the genes in the genome, which amounts to 50% of Atlantic salmon genes mapped to reactions in KEGG [11], and its focus is on core metabolism generating energy and biomass. This covers pathways that connect feed to fillet, which is a primary focus of research and aquaculture, but obviously excludes many other interesting processes such as synthesis of long-chain polyunsaturated fatty acids. Still, SALARECON performs very well according to all of our metrics: it is more similar to the latest zebrafish model [17] than to any other multicellular

eukaryote for which a model is available in BiGG [26, 29–31], achieves a MEMOTE score of 96%, which is better than all models in BiGG [10] (although many BiGG models could presumably be annotated and curated to reach a comparable score with reasonable effort), and performs all metabolic tasks within the scope of the model (amino acid, nucleotide, and energy metabolism). It also correctly classifies amino acids as essential [27, 36] and captures basic fish physiology, e.g. aerobic growth with uptake of essential amino acids, choline, and phosphate, and secretion of carbon dioxide and ammonia.

The extensive annotation of genes, metabolites, and reactions is a key strength of SALARECON that facilitates use with existing models, tools, and data. In particular, identifiers from BiGG [10] make it easy to compare and combine SALARECON with state-of-the-art models [59, 60], e.g. to predict interactions between Atlantic salmon and its gut microbiota. It also allows direct application of implemented methods such as evaluation of metabolic tasks [35]. The salmon-specific biomass reaction enables prediction of growth and related fluxes and is based on organism-specific data [27], making SALARECON a more realistic representation of salmon metabolism than a network reconstruction [12]. As demonstrated for Atlantic cod [18], even getting to this stage is challenging for non-model animals.

Our analysis of growth under oxygen limitation shows that phenotypes predicted by SALARECON can be fit to experimental data and produce detailed mechanistic explanations of Atlantic salmon physiology. Specifically, SALARECON explained hypoxic metabolism and growth in terms of metabolic fluxes with implications for fish welfare and productivity in aquaculture. The growth predictions depend on unknown environmental conditions and flux capacities, but SALARECON can be used to account for such uncertainty through random sampling. Average growth predictions from SALARECON fit the available data [41–44] as well as simple growth models and gave accurate estimates of critical water oxygen saturations in agreement with observations [52]. The predicted metabolic fluxes defined clusters of reactions with distinct pathway enrichments and contributions to hypoxic growth, notably suggesting that energy generation from glucose becomes saturated at low oxygen levels and that amino and fatty acids become more important energy sources with increasing oxygen. Predictions contrasting growth-limiting amino acids in three commercial feed ingredients also agreed well with data [55, 56] and showed that SALARECON can be used to evaluate the efficiency of sustainable feeds, a key challenge for modern aquaculture. Feed efficiencies predicted by SALARECON lie within reported ranges [53, 54] and suggest that the feed efficiency of fish meal can be achieved by supplementing one amino acid for insect meal and three for soybean meal. This shows that SALARECON can be used to evaluate both current and novel feeds, potentially reducing the need for expensive fish experiments *in vitro* or *in vivo*.

In future work, we will expand SALARECON to cover more processes such as lipid and carbohydrate metabolism in full detail, and we will tailor it to gut, liver, muscle, and other tissues using omics data and metabolic tasks [35]. We will also leverage automated metabolic reconstruction tools for microbes to build models of the Atlantic salmon gut microbiota [59]. By coupling tissue-specific models to each other and to gut microbiota models, we can make detailed and partially dynamic whole-body models [61]. This would be a major leap from available dynamic models [62] and provide a mechanistic alternative to state-of-the-art bioenergetics models [63], opening up new possibilities for understanding fish physiology and rational engineering of feeds, conditions, and genetics.

## Conclusion

SALARECON covers half of the annotated metabolic genes in the Atlantic salmon genome and can predict metabolic fluxes and growth with a salmon-specific biomass reaction. It has

been extensively annotated, curated, and evaluated, and it can be used to tackle research questions from fish physiology to aquaculture. In particular, SALARECON is a promising new tool for predicting breeding strategies and novel feeds that optimize for production parameters such as feed efficiency and impact on fish health and environment. Future work will expand SALARECON and integrate it with omics data to make tissue-specific and partially dynamic whole-body models. SALARECON should facilitate systems biology studies of Atlantic salmon and other salmonids, and we hope that it will be widely used by modelers as well as biologists.

## Supporting information

**S1 Fig. Draft model construction.** Flowchart showing the procedure used to add new pathways to the draft model or edit pathways already in the draft model. Pathways were added or edited one by one with information about reactions obtained from databases and literature. After adding or editing a pathway, the energy and redox balances and topological properties of the model, e.g. flux consistency, were checked. Based on the results from these analyses, the pathway was either kept or modified. Before final acceptance of a pathway, FBA was performed to ensure that the model was able to predict growth and metabolic fluxes.
(TIFF)

**S2 Fig. Adding nutritional supplements to a feed uptake reaction.** Feed uptake reactions are similar to biomass reactions, but supply metabolites rather than consuming them. The ratios between feed components are represented stoichiometrically, and scaled to sum to 1 g feed per mol uptake, so that one gram of the feed in the figure is equivalent to 2 mol A, 3 mol B and 1.4 mol C. With a fixed growth rate, the minimization of feed uptake is used as the objective of FBA. Surplus of metabolites in the feed uptake reactions are allowed to be exported via exchange reactions to avoid blocking the feed uptake reaction. Limiting metabolites can be identified from the reduced costs of the FBA solution. To avoid large molecules being favored, the reduced cost should be multiplied by the molecular mass ($M$) of the metabolite. Other factors such as price, $CO_2$ equivalents, or environmental cost could be taken into account in this step. The boundaries of the limiting exchange reaction are reversed to allow uptake, and the reaction is scaled by molecular mass and added to the objective. In this case, the cost of supplements is assumed to be equivalent to mass, but the cost could also be set to be higher than the other feed ingredients, which could be more realistic.
(TIFF)

**S3 Fig. Model degree distributions.** (a) Distribution of number of metabolites converted by reactions. Boundary reactions exchange one metabolite with the extracellular environment and transport reactions usually exchange an even number of metabolites between compartments. (b) Distribution of number of genes associated with reactions. Transport and boundary reactions lack annotation and are not associated with any genes. Most metabolic reactions (95%) are associated with one or more genes. (c) Cumulative distribution of number of reactions associated with genes and metabolites (number of genes or metabolites associated with $k$ or more reactions for all $k$). Most genes and metabolites are associated with a few reactions but some metabolites are highly connected hubs. Power law fits are shown for genes and metabolites.
(TIFF)

**S4 Fig. Dendrograms for models of multicellular eukaryotes.** Dendrograms from agglomerative hierarchical clustering of reaction contents of metabolic models of *Salmo salar* (SS), *Danio rerio* (DR) [17], *Mus musculus* (MM) [29], *Cricetulus griseus* (CG) [30], *Homo sapiens* (HS) [26], and *Phaeodactylum tricornutum* (PT) [31]. We combined 16 different dissimilarity

measures with five different clustering methods and computed the cophenetic correlation coefficient (CCC) [33] for each measure and method. SS and DR are highlighted in red.
(TIFF)

**S5 Fig. Reaction contents of models of multicellular eukaryotes.** Reaction contents of metabolic models of *Salmo salar*, *Danio rerio* [17], *Mus musculus* [29], *Cricetulus griseus* [30], *Homo sapiens* [26], and *Phaeodactylum tricornutum* [31]. Each row is an organism, each column is a reaction, and a dark cell indicates a reaction that is found in the model of that organism. Rows are clustered by Jaccard distance using the "average" method and the number of reactions is given for each organism.
(TIFF)

**S6 Fig. Metabolic task results by subsystem.** Ability of SALARECON to perform metabolic tasks [35]. Tasks are grouped by metabolic subsystem and classified as successful if model predictions reflected expected metabolic (in)capabilities.
(TIFF)

**S7 Fig. Conditions and growth rates from oxygen-limited growth analysis.** (a) Feed coefficients of amino acids and choline in conditions used to predict oxygen-limited growth (1,000 samples). The coefficients were randomly sampled from a uniform distribution. (b) Pairwise Pearson correlations between metabolites of feed coefficents shown in a. (c) Flux bounds for conditions used to predict oxygen-limited growth (1,000 samples). Flux bounds were randomly sampled from a lognormal distribution. (d) Pairwise Pearson correlations of flux bounds shown in a. (e) Predicted absolute growth rates as a function of absolute oxygen uptake rates for the 1,000 randomly sampled conditions. The absolute growth rates were not intended to be realistic and only relative growth rates were used in the analysis (normalized by maximum growth rate without oxygen limitation).
(TIFF)

**S8 Fig. Secreted metabolites in oxygen-limited growth analysis.** Secretion flux relative to growth rate from oxygen-limited growth simulations. Fluxes are shown for $CO_2$, $NH_3$, urea, and all other secreted metabolites combined. Mean relative flux across 1,000 randomly sampled conditions is shown with 95% confidence bands from bootstrapping with 1,000 samples.
(TIFF)

**S9 Fig. Effect of sampling on results from oxygen-limited growth analysis.** Results from oxygen-limited growth analysis with (a–c) 1,000 randomly sampled feeds and flux bounds, (d–f) 100 randomly sampled feeds with default flux bounds, and (g–i) a fish meal feed (Table 1) with default flux bounds. See legend for Fig 4a–4c.
(TIFF)

**S10 Fig. Effect of sampling on limiting reactions in oxygen-limited growth analysis.** Limiting reactions in oxygen-limited growth analysis with (a) 1,000 randomly sampled feeds and flux bounds, (b) 100 randomly sampled feeds with default flux bounds, and (c) fish meal feed (Table 1) with default flux bounds. Rows are reactions, columns are flux distributions sorted by condition and relative oxygen uptake rate, and a dark cell indicates that a reaction is limiting in a solution (i.e. has flux equal to a non-zero flux bound). Rows are clustered by Euclidean distance using Ward's minimum variance method.
(TIFF)

**S11 Fig. Growth-limiting amino acids in commercial feed ingredients.** Feed efficiency as a function of number of supplemented amino acids, measured in mg feed ingredient and

supplemented amino acids consumed / gDW biomass produced for (a) fish meal, (b) soybean meal, and (c) black soldier fly larvae meal. Amino acids are indicated by color and ordered from most limiting (left) to least limiting (right). Each bar represents the fed amount of amino acid sources, with one amino acid supplemented per step towards the right. Limiting amino acids were supplemented until all feed protein had been replaced.
(TIFF)

## Acknowledgments

We thank Ingunn Verne Ruud, Pedro Febrer Martínez, and Håvard Molversmyr for helpful contributions to the analyses and Snorre Sulheim for feedback on the manuscript.

## Author Contributions

**Conceptualization:** Liv Torunn Mydland, Arne B. Gjuvsland, Simen R. Sandve, Vitor A. P. Martins dos Santos, Jon Olav Vik.

**Data curation:** Maksim Zakhartsev, Filip Rotnes, Marie Gulla, Ove Øyås, Jesse C. J. van Dam, Fabian Grammes, Wout van Helvoirt, Jasper J. Koehorst, Arne B. Gjuvsland, Jon Olav Vik.

**Formal analysis:** Maksim Zakhartsev, Filip Rotnes, Marie Gulla, Ove Øyås, Fabian Grammes, Arne B. Gjuvsland, Jon Olav Vik.

**Funding acquisition:** Jon Olav Vik.

**Investigation:** Maksim Zakhartsev, Filip Rotnes, Marie Gulla, Ove Øyås, Fabian Grammes, Yang Jin, Liv Torunn Mydland, Arne B. Gjuvsland, Jon Olav Vik.

**Methodology:** Maksim Zakhartsev, Filip Rotnes, Marie Gulla, Ove Øyås, Jesse C. J. van Dam, Maria Suarez-Diez, Róbert Anton Hafþórsson, Wout van Helvoirt, Jasper J. Koehorst, Peter J. Schaap, Arne B. Gjuvsland, Jon Olav Vik.

**Project administration:** Jon Olav Vik.

**Software:** Filip Rotnes, Marie Gulla, Ove Øyås, Jesse C. J. van Dam, Róbert Anton Hafþórsson, Wout van Helvoirt, Jasper J. Koehorst, Arne B. Gjuvsland, Jon Olav Vik.

**Supervision:** Maria Suarez-Diez, Peter J. Schaap, Jon Olav Vik.

**Validation:** Maksim Zakhartsev, Filip Rotnes, Marie Gulla, Ove Øyås, Jon Olav Vik.

**Visualization:** Filip Rotnes, Ove Øyås, Jon Olav Vik.

**Writing – original draft:** Maksim Zakhartsev, Filip Rotnes, Ove Øyås, Jon Olav Vik.

**Writing – review & editing:** Filip Rotnes, Ove Øyås, Jesse C. J. van Dam, Maria Suarez-Diez, Fabian Grammes, Jasper J. Koehorst, Arne B. Gjuvsland, Simen R. Sandve, Jon Olav Vik.

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
