## [Decision Letter · Decision Letter 0]

19 Nov 2021

Dear Dr. Øyås,

Thank you very much for submitting your manuscript "SALARECON connects the Atlantic salmon genome to growth and feed efficiency" for consideration at PLOS Computational Biology.

As with all papers reviewed by the journal, your manuscript was reviewed by members of the editorial board and by several independent reviewers. In light of the reviews (below this email), we would like to invite the resubmission of a significantly-revised version that takes into account the reviewers' comments.

All reviewers unanimously raised concerns about the lack of sufficient details and quite critical about the potential physiological concerns existing in the model. Thus, it would be absolutely crucial to improve the clarity of the manuscript and provide the requested details including the results of simulations asked by reviewers. Furthermore, it is important that the model and the process of curation is available to everyone using mentioned version control according to the best practices in the community.  

We cannot make any decision about publication until we have seen the revised manuscript and your response to the reviewers' comments. Your revised manuscript is also likely to be sent to reviewers for further evaluation.

Sincerely,

Aleksej Zelezniak

Guest Editor

PLOS Computational Biology

Kiran Patil

Deputy Editor

PLOS Computational Biology

All reviewers unanimously report the lack of sufficient details and quite critical concerns about the model, the manuscript cannot proceed to publication unless all comments are sufficiently addressed and supported.

Reviewer's Responses to Questions

**Comments to the Authors:**

Reviewer #1: Salmon is indeed a commercially very valuable fish in the market and it has high global demand. Therefore, in order to meet the global demand, it is produced at a higher scale. However, the challenge is to get the feed correct (similar to the level of marine feed) so that good fish quality is maintained. In this work, the authors have developed the genome-scale metabolic model of salmon fish and then used the model as a tool to examine a different type of commercial fish feed. The model predictions with regards to different feed and supplementation of few amino acids are in line with other experimental observations. Furthermore, the model is capable of mimicking hypoxic growth. Overall, the procedures followed to obtain the model are standard. I believe that the model will be used as a valuable tool to study salmon metabolism and growth.

In my view, the manuscript (the main text) is written very well but the quality of the figures is very poor. In most of the figures, I cannot read what is written in x and y-axis labels. Other texts on the figures are unclear, therefore, I am afraid, it is difficult to judge the quality of the paper. I am happy to go through the paper once the good-quality figures are provided.

Other than figures (results), which need additional review, I have few more comments.

Line 47-50: Authors mention that due to lack of information, GPR is converted to OR type for all reactions. Is it normal to do so? I believe such detailed experimental information is not available in zebrafish as well as other higher organisms, so how do those models keep GPR relations?

Analysis of oxygen-limited growth is not clear to me. I did not understand what is 100 randomized conditions? Is it 100 randomly chosen nutrients from the list of all exchange metabolites? Or do you have a media setup and then you chose the reaction bound randomly 100 times? In minimal feed, authors mention amino acids and choline. Which amino acids are essential here?

Line 117: Authors minimize the feed uptake. In my opinion, it needs a more detailed explanation. Is it one of the objective function reactions where all feed constituents are lumped into a feed reaction, similar to one in the biomass? Or do you minimize uptake reactions one by one iteratively while keeping the rest constant? I did not fully understand this part.

Line 129: I am surprised to see this low percentage of metabolic gene coverage in the model. Only 48% of kegg metabolic genes are covered in the model. Why are the rest 52% of metabolic genes from kegg not included? I think this is a very big percent of genes to ignore -- and they are mostly metabolic genes as given in the KEGG. I am not saying all kegg metabolic genes should be included but leaving 52 % is too much I think. I think this low coverage of metabolic genes is the main reason why the model achieved only 53% MEMOTE score. If the aim is to understand the specific tasks so, in that case, can we not say that the reconstructed model is a core model considering only central pathways/reactions?

Reviewer #2: The authors have 1) reconstructed a metabolic model of Atlantic salmon, 2) conducted a number of tests of the model both standardized and custom, the later including its Jaccard distance to other models; essentiality of amino acids; relative growth rate with randomly sampled nutrient uptake and flux capacities, 3) used the model to evaluate the effect of amino acid supplements to different feeds. While new metabolic models are always useful and needed, the paper seems too limited in biological insight and in motivation of its methodology, as detailed in the following specific comments.

Major comments

The authors interpret the Jaccard distance as phylogenic distance. However, other factors may also influence the metric, e.g. the size or focus of the reconstruction, in figure S2 it is reported that the different models span a range of 456-3554 reactions.

The model constitutes a unique opportunity to compare Salmon metabolism to that of other organisms, but no fish specific biological conclusions seem to be drawn, e.g. differences in nitrogen excretion or oxygen utilization. It is not clear how the results would differ if a model from another organism was used, assuming that the biomass equation was amended. Also differences in arginine metabolism between fish and mammals would be interesting to discuss, arginine is only considered a conditionally essential amino acid in human. The reference cited for its essentiality in salmon does not seem to be the original source of the claim.

The method to randomly sample input and flux constraints appears to be novel since no references are given, but it is not thoroughly motivated, described, or analyzed, e.g. what would the effect be of only sampling inputs or constraints? Is it reasonable to assume that only essential amino acids are provided and that they are uniformly distributed? How is the sampling of constraints motivated? How is this approach expected to fair with respect to well-known biases from sampling of highly interdependent fluxes (Heinonen 2019). The results of this method would require careful study, to rule out unintended effects of the imposed constraints, e.g. in the analysis of the flux distributions the flux through glycolysis is implicitly assumed to be energy-generating, however, since uptake of glucose is blocked, and glycogen is in the biomass equation, the flux is likely reversed (gluconeogenesis). The manuscript mentions blockage of nutrient secretion of feed nutrients, does this mean that the constraint only applies to the amino acids, what would be the consequence of not doing so?

A set of fitting-parameters and relative rates are used to fit the model to experimental data, this, however, seems unnecessary, since the minimum oxygen uptake should follow from the oxygen uptake required for maintenance ATP expenditure and the maximum oxygen uptake should depend on the maximum specific growth rate together with the growth associated ATP expenditure, i.e. more explanatory parameters that could be estimated from the data.

Fish meal is used as reference in Figure 5, but presumably a similar conclusion could more easily be drawn by comparing the optimal amino acid composition identified by the model to the to the composition in the feed. Since only essential amino acids are considered, the interconversion capacity of the model will likely not be utilized, and the optimal composition is therefore presumably highly similar to the amino acid composition in biomass, in particular since unlimited glucose is supplied so that the possibility for ATP constrained growth is neglected. This, btw, does seem to be a reasonable assumption under optimized growth conditions (El-Mowafi 2010). On a more general level the feed optimization assumes no effects of gut microbiota or obligate losses, which perhaps should be discussed.

Minor comments

There seems to be no reason to use the arbitrary specific growth rate of 1 h-1, which is unphysiologically high. Similar to elsewhere in the manuscript yields, gdw/gdw/mmol O2 could be reported.

Figure 4d could be improved for interpretability. How sensitive are the clusters to the 100 random samples, if 1000 samples were used, would the figure look the same?

Due to differences in specific activity between enzymes, minimized flux is not the same as minimal enzymatic cost (line 196).

On line 129 it is stated that the model focuses on energy metabolism, but on line 279 it is stated that it focuses on amino acid, nucleotide, and energy metabolism. This a more accurate description, since most of the results relate to amino acids.

It is unusual to see the Monod model with an intercept term as in figure 4b.

References

Markus Heinonen, Maria Osmala, Henrik Mannerström, Janne Wallenius, Samuel Kaski, Juho Rousu, Harri Lähdesmäki, Bayesian metabolic flux analysis reveals intracellular flux couplings, Bioinformatics, Volume 35, Issue 14, July 2019, Pages i548–i557

El-Mowafi, A., Ruohonen, K., Hevrøy, E.M. and Espe, M. (2010), Impact of digestible energy levels at three different dietary amino acid levels on growth performance and protein accretion in Atlantic salmon. Aquaculture Research, 41: 373-384.

Reviewer #3: The authors present a metabolic model of atlantic salmon. They compare the model to genome-scale models of other organisms, and use the model for looking into areas of metabolism that change under varying feed/oxygen uptakes, and detecting specific amino acids that are limiting in different feed sources. Overall the paper is well written and is a good foundation stone for salmon modeling work to come. I commend the authors for using several state of the art practices in model development, such as unit testing and memote validation. However, there are a few major points that should be addressed in the manuscript and model before publication.

Major comments:

* L37: The description of how the draft model was generated is extremely brief; a more thorough explanation of the methods used is required, especially considering that a) the most important result of this manuscript is the model itself, and b) the authors used proprietary software for it, which hinders reproducibility. Was a template model used to build this reconstruction, or was it built from scratch? Which alignment method was use to map the genes to KEGG, or was that information directly taken from the annotated genome? Which criteria were used for adding reactions to the draft model? Which criteria were used to finish adding reactions? Was there any gap-filling performed? How did the authors ensure the model could produce biomass?

* L47: Assuming every gene-reaction-relationship is an OR relationship is a quite strong assumption, that will lead to many false negatives when using the model for gene essentiality predictions (one of the most common uses for metabolic models). Most reconstruction tools are able to account for complexes; if insilico discovery cannot do it, I would suggest, for instance, using a reference model for retrieving this information: if the same reaction is present in both models, and there is homology between the corresponding genes, the AND/OR relationship from the reference can be copied.

* L78: This section reads quite confusing and it took me a few reads to understand what the authors did. Considering that most of the result section is about the oxygen simulations, the authors should either make this section clearer to read, or remove focus from it to highlight the (much clearer) next section "Predicting growth-limiting amino acids in feeds".

* L141: It is misleading to claim that this model accounts for the metabolic production of lipids, as the only lipid accounted for in the biomass reaction is PC 16:0/16:0, i.e. most of the rich lipid metabolism that fish has is ignored in the model, and the uptake costs of choline are largely over-estimated. Similarly, carbohydrates are only represented with glycogen. Considering the many published semi-automated tools for model generation, have the authors considered adding some additional metabolic content to their model so that it can produce some additional biomass constituents, e.g. lipids and/or carbohydrates, but also vitamins, cofactors, trace minerals? If not, then at the very least the authors should consider removing mentions to those pathways as being fully accounted for in the model.

* L163: Only at this point in the manuscript I understood the scope of the model (energy, amino acid, and nucleotide metabolism). I believe the abstract, introduction and any other summary section of the manuscript should clearly highlight this scope and mention that this is a smaller-scale / highly-curated model and not genome-scale model, otherwise it reads misleading, making the reader think that this model is part of the genome-scale family. Furthermore, the authors should provide a richer discussion about what are the advantages/disadvantages of generating such a model instead of using the (more prevalent) full genome-scale model approach.

* The model repository is quite unorganized and could benefit from some cleanup: I would recommend following a standard (e.g. https://github.com/SysBioChalmers/Human-GEM or the broader https://github.com/drivendata/cookiecutter-data-science). Specifically, the following ideas would increase readability of the repo:

* Group scripts/notebooks in a separate folder.

* Store the S. salar model in a simpler path than models/sasa/salarecon_bigg_curated.xml

* It is confusing to find the model within a plethora of genome-scale models from different species. Furthermore, is it even needed to store models from other studies in this repo? Providing the links from BiGG should be enough.

* Having a folder with old versions of the model defeats the purpose of using git: Shouldn't those models be accessible through specific commits in the git history? Labels can be provided to those commits for quick reference.

Minor comments:

* L39: Please detail (in methods or in supplementary material) the settings that were chosen for WoLF PSORT and SAPP.

* L44: Unclear to me what "We iteratively converted the model to the BiGG namespace" means: why would the authors need to convert the model ids more than once to BiGG?

* L56: It is unclear to me what was the exact criteria for stopping model annotation/curation.

* L80: For a randomized approach like the one the authors present, 100 randomized conditions is probably not enough for properly sampling the solution space. I would recommend the authors increasing this number to at least 1000. On the other hand, 100 linearly spaced oxygen uptake rates is probably redundant and could be replaced with a smaller number, considering that the response of the model seems to be more or less linear (Fig. 4d).

* L89: Here I am still confused: as far as I understand there were 100 (feed random uptake combinations) * 100 (O2 linearly spaced O2 uptakes) = 10,000 simulations performed. Are there also other reaction bounds randomly sampled?

* L133: If the authors claim that the node degree distributions of the network "are typical for metabolic and other biological networks", then I would expect Figure S1 to show numbers for those networks, however only the information for the salmon model is presented.

* Figures 3a, 3d and 4c are not referenced anywhere in the manuscript as far as I can tell.

* Fig. 3d: Amino acid essentiality data should be shown together with the predictions here, otherwise the statement in the caption "Essential amino acids predicted by SALARECON match data" is not proven.

* L205: When performing pathway enrichment, it is standard practice to show in the plot (Fig. 4f) the number of reactions that each pathway contains, as a pathway with less than e.g. 5 reactions in the model should probably be filtered out, given that already a couple of reactions showing in a cluster could be counted as significantly over-represented. Note that the filtering here should be done by reactions, not genes (to not account for the same flux value multiple times).

* L263: "Automatically built models work well for microbes but are still outperformed by models that are built by manual iteration" This is a strong claim that I believe requires at least some references and explaining what is meant by "outperforming".

* L277: Comparing the memote score of this model to models in BiGG is in my opinion unfair, as all those models were published before the memote score was introduced, and therefore did not account for it in their development. Furthermore, by adding a few things to those models, such as annotation and/or SBO terms, many of those models would increase their score to close to 95% as well.

* L295: This is achieved only because the biomass reaction is relatively simple (e.g. one type of carbohydrate and one type of PC), so I'm not sure it should be highlighted as a great advantage of the model.

* L320: Check grammar of sentence.

* L356: Should say S3 Fig.

**Have the authors made all data and (if applicable) computational code underlying the findings in their manuscript fully available?**

Reviewer #1: Yes

Reviewer #2: Yes

Reviewer #3: Yes

PLOS authors have the option to publish the peer review history of their article (what does this mean?). If published, this will include your full peer review and any attached files.

Reviewer #1: **Yes: **Dr Mohammad Tauqeer Alam, Department of Biology, United Arab Emirates University, UAE

Reviewer #2: No

Reviewer #3: **Yes: **Benjamín J. Sánchez
---

## [Decision Letter · Decision Letter 1]

6 Apr 2022

Dear Dr. Øyås,

Thank you very much for submitting your manuscript "SALARECON connects the Atlantic salmon genome to growth and feed efficiency" for consideration at PLOS Computational Biology. As with all papers reviewed by the journal, your manuscript was reviewed by members of the editorial board and by several independent reviewers. The reviewers appreciated the attention to an important topic. Based on the reviews, we are likely considering this manuscript for publication, providing that you address all reviewers comments.

Sincerely,

Aleksej Zelezniak

Guest Editor

PLOS Computational Biology

Kiran Patil

Deputy Editor

PLOS Computational Biology

[LINK]

Reviewer's Responses to Questions

**Comments to the Authors:**

Reviewer #1: I have gone through the comments and rebuttal. All of my concerns were addressed by the reviewers. They have revised the manuscript accordingly, and I have no further comments. Brilliant job! Congratulations to the authors for a nice piece of work.

Reviewer #2: For this revision of their paper on metabolic modeling of Atlantic salmon the authors have developed the description of their methodology and have expanded the biological analysis. However, some questions remain to be addressed as detailed in the following specific comments.

Major comments

1. The authors maintain that Jacard distance is indicative of similarity between fish metabolism and a metric of salmon-specificity, “The models clustered by phylogeny with fish and mammals forming distinct groups and the diatom as an outlier, indicating that SALARECON captures fish- and likely salmon-specific metabolism.”. If these models were all genome-scale reconstructions, this may be a correct conclusion, however, since many reactions are missing from SALARECON due to its limited scope, it is not apparent if the clustering is driven by similarity in scope or in content. As an illustration: if every reaction in the salmon reconstruction is also present in the human reconstruction, the Jaccard distance would still be very large ~0.85 (1 - 456/3554), due to difference in reconstruction size, while the corresponding value for zebra fish would be 0.65, and in such case they would thus cluster due to similarity in scope. At best the results are consistent with the hypothesis that SALARECON captures fish and salmon-specific metabolism, but Jaccard distance does simply not seem like an appropriate metric when comparing reconstructions of different scope.

2. Similarly, in Figure 2e it is not clear if the observed differences are due to difference in species or reconstruction size, all differences seem to suggest that the Human model has additional capacity, e.g. for the human model, the phosphate likely originates from phospholipid metabolism, which is not reconstructed in SALARECON. It is not clear if SALARECON has any metabolites or reactions that are not also present in the human reconstruction, and it is not clear if the reactions or metabolites that are missing in SALARECON are due to limited reconstruction scope or genuinely missing.

3. The Monod model with an intercept term is not the Monod model. It is perhaps an “Extended Monod model”. However, I the r/r _max term was used instead of x in the Monod equation, there would likely be no need for an intercept and the results would be more comparable to the model as they receive the same input.

4. Regarding the choice of uniform distribution for nutrient sampling, the Authors write in their point-by-point response “we do not have prior information and therefore choose to sample from a uniform distribution”, however, is not the fish meal in Fig.5a such prior knowledge?

5. The capacity constraints are sampled from a log normal distribution, spanning 6 orders of magnitude. How do the authors ensure that the uptake constrains are of a corresponding magnitude? The manuscript states that uptake is normalized to 1 g/gdw/h, and that this value is arbitrary, however, it will not be arbitrary if the flux capacities are constrained, e.g. if uptake is much larger than capacity, then capacity will determine growth, and if uptake rates are much lower than capacity, then uptake will determine growth. To ensure that both constraints are active the authors could compare the curve in Figure 4a with and without uptake constraints. Alternatively, the predicted fluxes could be compared with the constraints to determine the frequency of constraining growth (flux==constraint) for each internal- and uptake reaction.

6. The revised sampling method resembles the one used by Beg et al 2007, further described by Adadi et al 2012, it may be worth checking if they are mathematically the same as the proposed method.

7. The authors state that the purpose of figure 4 is to “show that SALARECON can give good estimates of physiological parameters (minimal and maximal water oxygen saturation)”. However, the minimal water oxygen saturation is a fitting parameter (x0) and the maximum is another fitting parameter (x1), so it is not clear that the results achieve this purpose.

Minor comments:

1. The definition of Jaccard distance is in equation 1 seems to be the Jaccard similarity, for distance: 1 – J(A, B).

2. It may be contested that Robinson et al. 2020 have reconstructed a more recent human model than Recon3D, but perhaps this is the most recent model in the BIGG database.

3. The authors sample the upper- and lower bounds of reversible fluxes independently. Since the [E] term is present in both the forward and backward reaction, they are in principle not independent. However, in practice this will not matter since pFBA ensures that each reaction will be either in the forward or backward direction.

4. Figure 3c says “fitted value” on y axis, but R2 is the calculated coefficient of determination, not a fitted value.

5. Figure S6F shows absolute growth rate and uptake rates. The growth rate is 1% per hour, which is approximately 24 times higher than observed in salmon Cook et al 2000.

6. In figure 2e the figure legend could perhaps be more clearly state that slash (“/”) indicates sets of metabolites, e.g. NH3 is present in both the red and blue box, but this is presumably because it is compared against the whole set NH3/Urea/Urate.

References

Beg, Q. K. et al. Intracellular crowding defines the mode and sequence of substrate uptake by Escherichia coli and constrains its metabolic activity. Proc. Natl. Acad. Sci. U. S. A. 104, 12663–12668 (2007).

Adadi, R., Volkmer, B., Milo, R., Heinemann, M. & Shlomi, T. Prediction of Microbial Growth Rate versus Biomass Yield by a Metabolic Network with Kinetic Parameters. PLoS Comput. Biol. 8, e1002575 (2012).

Robinson, J et al. An atlas of human metabolism. Sci. Signal. 13, eaaz1482 (2020).

Adadi, R., Volkmer, B., Milo, R., Heinemann, M. & Shlomi, T. Prediction of Microbial Growth Rate versus Biomass Yield by a Metabolic Network with Kinetic Parameters. PLoS Comput. Biol. 8, e1002575 (2012).

Cook, J. T., McNiven, M. A., Richardson, G. F. & Sutterlin, A. M. Growth rate, body composition and feed digestibility/conversion of growth-enhanced transgenic Atlantic salmon (Salmo salar). Aquaculture 188, 15–32 (2000).

**Have the authors made all data and (if applicable) computational code underlying the findings in their manuscript fully available?**

Reviewer #1: Yes

Reviewer #2: Yes

PLOS authors have the option to publish the peer review history of their article (what does this mean?). If published, this will include your full peer review and any attached files.

Reviewer #1: **Yes: **Mohammad Tauqeer Alam, Department of Biology, United Arab Emirates University, Al Ain, Abu Dhabi.

Reviewer #2: No

Figure Files:

Data Requirements:

Reproducibility:

References:

---

## [Editor Report · Decision Letter 2]

10 May 2022

Dear Dr. Øyås,

We are pleased to inform you that your manuscript 'SALARECON connects the Atlantic salmon genome to growth and feed efficiency' has been provisionally accepted for publication in PLOS Computational Biology.

Best regards,

Aleksej Zelezniak

Guest Editor

PLOS Computational Biology

Kiran Patil

Deputy Editor

PLOS Computational Biology

---

## [Editor Report · Acceptance letter]

7 Jun 2022

PCOMPBIOL-D-21-01635R2 

SALARECON connects the Atlantic salmon genome to growth and feed efficiency

Dear Dr Øyås,

I am pleased to inform you that your manuscript has been formally accepted for publication in PLOS Computational Biology. Your manuscript is now with our production department and you will be notified of the publication date in due course.

With kind regards,

Agnes Pap
